PREPARED FOR SUBMISSION TO JHEP

# Bayesian Active Search on Parameter Space: a 95 GeV Spin-0 Resonance in the $(B - L)$SSM

**Mauricio A. Diaz**[a], **Giorgio Cerro**[a], **Srinandan Dasmahapatra**[b], **Stefano Moretti**[a,c]

[a] *School of Physics & Astronomy, University of Southampton, Southampton SO17 1BJ, UK*

[b] *School of Electronics & Computer Science, University of Southampton, Southampton SO17 1BJ, UK*

[c] *Department of Physics & Astronomy, Uppsala University, Box 516, 75120 Uppsala, Sweden*

*E-mail:* m.j.ardiles-diaz@soton.ac.uk; g.cerro@soton.ac.uk; sd@ecs.soton.ac.uk; stefano@soton.ac.uk; stefano.moretti@physics.uu.se

ABSTRACT: In the attempt to explain possible data anomalies from collider experiments in terms of New Physics (NP) models, computationally expensive scans over their parameter spaces are typically required in order to match theoretical predictions to experimental observations. Under the assumption that anomalies seen at a mass of about 95 GeV by the Large Electron-Positron (LEP) and Large Hadron Collider (LHC) experiments correspond to a NP signal, which we attempt to interpret as a spin-0 resonance in the $(B - L)$ Supersymmetric Standard Model $((B - L)$SSM), chosen as an illustrative example, we introduce a novel Machine Learning (ML) approach based on a multi-objective active search method, called b-CASTOR, able to achieve high sample efficiency and diversity, due to the use of probabilistic surrogate models and a volume based search policy, outperforming competing algorithms, such as those based on Markov-Chain Monte Carlo (MCMC) methods.

# 1 Introduction

The Standard Model (SM) of particle physics was finally confirmed in all its sectors after the discovery of a Higgs boson with a mass of 125 GeV at the Large Hadron Collider (LHC) in July 2012 [1, 2]. While the SM has been successful in explaining the elementary constituents of matter and their interactions, several hints of New Physics (NP), known as anomalies, are slowly arising [3, 4], bringing new research questions to the current status of particle physics. These anomalous results span over a large energy range, a brief review can be found in [5], coming from precision measurements and direct experimental searches, including flavour observable, anomalous magnetic moment of the muon, the $W^{\pm}$ boson mass and the possible existence of additional neutral spin-0 particle, the latter being the primary focus of this work. Anomalous experimental signals at $\approx 95 GeV$ have been reported in searches for new Higgs bosons. Different experimental analyses support this anomaly: a $\gamma\gamma$ (di-photon) excess seen at CMS [6], a $\tau^+\tau^-$ (di-tau) excess again by CMS [7] as well as a $b\bar{b}$ excess seen by LEP [8].

Such anomalies can be addressed by Beyond the Standard Model (BSM) scenarios. The latter are motivated by the need to explain, either individually or collectively, a variety of flaws of the SM: e.g., the absence of neutrino masses, the unexplained baryon-antibarion asymmetry in the universe, no candidate for Dark Matter (DM), etc. It is thus intriguing to see whether such theoretical constructions can also be used to explain the aforementioned 95 GeV anomalies. In fact, a variety of such BSM scenarios have been invoked in the latter context: [9–12]. Of all such theoretical frameworks, we focus here on the $(B-L)$SSM, as a distinctive example of a model realisation of Supersymmetry that can explain all such aforementioned SM flaws (see Ref. [13]) as well as the $\gamma\gamma$ anomaly at 95 GeV [14]. From a physics point of view, the purpose is thus to assess whether such a BSM scenario is also able to explain the $\tau^+\tau^-$ and $b\bar{b}$ excesses.

However, there is a pressing technical problem in such an endeavour: the regions in the parameter space of a BSM scenario capable of accommodating a combination of experimental results are sparse and possibly disconnected[1]. Furthermore, the parameter space of these models is typically highly dimensional and is defined within large ranges of the fundamental inputs. The computational cost associated with numerically evaluating a specific configuration of a BSM model using a typical High Energy Physics (HEP) software toolbox is high.

The HEP community has lately relied on Monte Carlo Markov Chain (MCMC) sampling methods to explore the parameter space of particle physics models: HEPfit [15], GAMBIT [16] and Magellan [17] are examples of public toolkits of this kind[2], developed to address the demand for readily applicable sampling strategies. Despite the effectiveness of MCMC methods (and variations thereof) to perform probabilistic inferences, fitting models to data and their widespread use in the physical sciences [19], these methods still confront the obstacles discussed above.

Continuous developments in ML based methods for exploring parameter spaces in BSM scenarios aim to address these challenges. Neural Network (NN) based methods [20, 21] have been proposed, adopting different formulations, such as *regression* and *classification*. For regression the physical observables are learned in an incremental manner, while for classification the viability of a parameter space configuration is treated as a label. Further, in both cases the learned model is incorporated into a policy to perform an informed sampling strategy. An alternative approach was developed in [22, 23], using Active Learning (AL) practices to train a NN discriminator. The primary aim of this

---

[1]In this work, the total region in the parameter space that can accommodate a combination of desired values for the objectives is referred to as the *Satisfactory* region and is denoted by $\mathcal{S}$.

[2]A modification to the MCMC approach very popular in the HEP community is 'nested sampling', as used in Multinest [18].

approach is to incrementally learn the decision boundary in regions of the parameter space where the model is allowed. Although NN based methods offer diversity in the search space configurations leading to an informative characterisation of the satisfactory regions, they require large datasets to achieve high accuracy, which can be challenging when dealing with computationally expensive HEP toolbox. This sample efficiency problem was noted in [24], where an alternative strategy was employed, re-framing a parameter space scan as a single-objective Black-Box Optimisation (BBO) problem. Although in single-objective optimisation there is a clear notion of optimality, many cases in BSM phenomenology involve multiple conflicting objectives, i.e., observables, to be optimised simultaneously. Multi-objective optimisation introduces the concept of Pareto front (also called Pareto frontier or Pareto curve), the set of solutions where no individual objective can be improved without loss in at least one other objective. However, for multi-objective optimisation problems [25], the availability of a broader set of solutions, those that are near optimal, allows for a more comprehensive characterisation and understanding of the black-box function, i.e., the BSM model under study. In the context of BSM phenomenology, a greater effective sample set correlates to the notion of fine-tuning, meaning that a large and diverse set of solutions in a model corresponds to a low fine tuning. Techniques such as AL [26] and Active Search (AS) [27] are employed to extend optimisation beyond the Pareto front.

This work aims at filling identified gaps in the existing literature by focusing on improving sample efficiency for computational expensive numerical evaluations of BSM models (the aforementioned $(B-L)$SSM being out benchmark example), a comprehensive characterisation of the total region in their parameter space that satisfy desired values in multiple observables while achieving sample diversity within these discovered regions. A batched Bayesian Multi-Objective (BMO) AS approach is developed which we name b-CASTOR[3]. In this approach, multiple phenomenological signatures of a particular BSM model are set as the multiple objectives, constrained by experimental measurements. These can refer to particle masses, Branching Ratios (BRs), production cross-sections or any model prediction information. To make parameter sampling more efficient, we use surrogate models to approximate the objectives, a common practice in Bayesian Optimisation (BO) [28]. In particular, we model the objectives with Gaussian Process based models[29]. This probabilistic formulation enables us to introduce the expected coverage improvement acquisition function, introduced in [27]. This policy seeks to improve volume coverage within the satisfactory regions of the search space to propose new candidate solutions in a sequential manner. Lastly, we develop a sampling strategy to allow multi-point evaluations and tunable control over the exploration and exploitation trade-off. This balances the extent to which parameters are selected far from the best predictions of the surrogate model that adapts to the previously chosen samples.

Two test experiments are presented, a double-objective 2D test function and a BSM phenomenology study. We compare the results of the proposed search method b-CASTOR with MCMC implementation [19, 30], based on the Metropolis-Hastings algorithm. As intimated, regarding the BSM phenomenology analysis, our focus is on the $(B-L)$SSM, a non-minimal realisation of Supersymmetry. It features a diverse Higgs sector composed of two Higgs doublets and two Higgs singlets, which can then be used to potentially explain (some of) the anomalies seen at $\sim 95$ GeV [31–33]. In addition, the $(B-L)$SSM introduces three extra neutralinos, corresponding to the Superpartners of two singlet scalars and a new neutral gauge boson $Z'$. The lightest neutralinos could function as potential candidates for cold Dark Matter (DM) [34]. Being a Supersymmetric scenario, it does not suffer from the hierarchy problem. Furthermore, it embeds a neutrino mass generation mechanism in the form of seesaw dynamics. Hence, it is a model which goes above and beyond the mere purpose of explaining data anomalies. Our findings indicate that the algorithm efficiently characterises the

---

[3]For **b**atched **C**onstraint **A**ctive **S**earch with **T**PE **O**ptimisation and **R**ank based sampling.

satisfactory region in the parameter space of the $(B-L)$SSM that can explain the LHC anomaly in the $h \to \gamma\gamma$ channel, enhancing scans conducted in previous studies [31–33]. However, given updated experimental results, it is essential to consider additional channels, such as $h \to \tau\tau$ (LHC) and $h \to b\bar{b}$ (LEP). With this updated experimental information, our approach did not find points capable of simultaneously explaining all three channels.

The plan of the paper is as follows. The next section is devoted to explain how the discussed experimental anomalies can potentially be linked to the chosen theoretical framework. Then, we introduce our MO AS approach. This is then followed by our results and conclusions, in turn.

## 2 Higgs Bosons in the $(B-L)$SSM

### 2.1 Experimental Searches

Results for new Higgs boson experimental searches are expressed as limits on a variable $\mu$, known as the *signal strength modifier*, that measures one parameter scalings of the total SM rate for a particular signal channel or ensemble of signal channels. These variables are defined as the production cross section for a specific channel times a decay BR involving the new Higgs boson, normalised to the SM value of the same process, for a given mass. In this work we start by focusing on the combined results of Higgs boson searches at CMS [6] and ATLAS [35] in the $\gamma\gamma$ final state. The searches reported excesses of $2.9\sigma$ and $1.7\sigma$ in the two experiments, respectively, with a (resonant) mass value of 95.4 GeV. The relevant signal strength mesured is expressed as follows:

$$\mu_{\gamma\gamma}^{\text{exp}} = \mu_{\gamma\gamma}^{\text{ATLAS+CMS}} = \frac{\sigma^{\text{exp}}(gg \to \phi \to \gamma\gamma)}{\sigma^{\text{SM}}(gg \to H \to \gamma\gamma)} = 0.27^{+0.10}_{-0.09}, \tag{2.1}$$

where $\phi$ is the possible particle behind the observed anomaly and $H$ is a would-be SM Higgs boson, both with a 95.4 GeV mass. This is the anomaly which was addressed within the $(B-L)$SSM in Ref. [31] and for which an explanation was found therein.

Additionally, though, two other search channels presented anomalies which support the possibility of such a $\gamma\gamma$ resonance. LEP [8] reported a now long-standing anomaly in searches for light Higgs bosons in the $e^+e^- \to Z(H \to b\bar{b})$ channel, corresponding to a $2.3\sigma$ local excess at a Higgs mass 98 $GeV$[4], leading to a signal strength modifier given by

$$\mu_{bb}^{\text{exp}} = \frac{\sigma\left(e^+e^- \to Z\phi \to Zb\bar{b}\right)}{\sigma^{SM}\left(e^+e^- \to ZH \to Zb\bar{b}\right)} = 0.117 \pm 0.057. \tag{2.2}$$

The CMS collaboration has detected an excess in the low-mass region for the gluon-fusion production mode and decay into $\tau^{\pm}\tau^-$ pairs [7], which is consistent with the excess observed in the di-photon search by CMS. For a mass value of 95GeV, CMS has reported a local significance of $2.6\sigma$. This corresponds to a signal strength

$$\mu_{\tau\tau}^{\text{exp}} = \frac{\sigma^{\text{exp}}\left(gg \to \phi \to \tau^+\tau^-\right)}{\sigma^{\text{SM}}\left(gg \to H \to \tau^+\tau^-\right)} = 1.2 \pm 0.5. \tag{2.3}$$

The reported results of searches for light neutral scalars at the LHC, using the $\gamma\gamma$ and $\tau^+\tau^-$ channels, and at LEP, using the $b\bar{b}$ channel, when combined, offer compelling evidence supporting the interpretation of these NP signals within the framework of BSM theories. This assumed scenario constitutes the focus of our investigation, which we carry out by first extending previous studies on a possible explanation of the $\gamma\gamma$ anomaly with the $(B-L)$SSM model and finally by considering all three results simultaneously.

---

[4]Note that the various mass values reported here are consistent with each other given the limited mass resolutions, especially in the case of $b\bar{b}$ and $\tau^+\tau^-$ final states.

## 2.2 The $(B - L)$SSM

The $(B - L)$SSM is essentially the Minimal Supersymmetric Standard Model (MSSM) extended by a $U(1)_{B-L}$ gauge symmetry,

$$\mathcal{G} = U(1)_Y \otimes SU(2)_L \otimes SU(3)_c \otimes U(1)_{B-L},$$

wherein the $U(1)_{B-L}$ symmetry (which is an accidental one in the SM) is spontaneously broken through the Higgs mechanism. The Superpotential of the model is given by

$$
\begin{aligned}
W_{\text{BLSSM}} =& y_u \hat{Q} \hat{H}_2 \hat{U}^c + y_d \hat{Q} \hat{H}_1 \hat{D}^c + y_e \hat{L} \hat{H}_1 \hat{E}^c + \mu \hat{H}_1 \hat{H}_2 \\
& + y_\nu \hat{L} \hat{H}_2 \hat{N}^c + y_N \hat{N}^c \hat{\chi}_1 \hat{N}^c + \mu' \hat{\chi}_1 \hat{\chi}_2 .
\end{aligned}
\tag{2.4}
$$

Here, the first four terms corresponds to the MSSM Superpotential, incorporating the Yukawa interactions with their respective Yukawa couplings, namely $y_u, y_d$ and $y_e$. Additionally, a bilinear term between the $SU(2)_L$ Higgs doublet Superfields $\hat{H}_1$ and $\hat{H}_2$ with opposite hypercharge $Y = \pm 1$ represents a globally Supersymmetric Higgs mass term. Additional terms describe the interactions between the (s)neutrinos $\hat{N}$ and the singlet Higgs Superfield $\hat{\chi}_1$. The corresponding Yukawa coupling constants are denoted as $y_v$ and $y_N$. Furthermore, $Q$ and $L$ denote the left-handed quark and lepton doublet Superfields, while $\hat{U}$, $\hat{D}$ and $\hat{E}$ represent the right-handed up-type, down-type and electron-type singlet Superfields, respectively. The corresponding soft Supersymmetry breaking terms and the details of the associated spectrum can be found in Refs. [33, 36, 37]. (The superscript $c$ represents charge conjugation.)

The masses for the physical neutral $(B - L)$SSM Higgs states can be obtained from

$$
\begin{aligned}
H_{1,2}^0 &= \frac{1}{\sqrt{2}} \left( v_{1,2} + \sigma_{1,2} + i \phi_{1,2} \right), \\
\chi_{1,2}^0 &= \frac{1}{\sqrt{2}} \left( v'_{1,2} + \sigma'_{1,2} + i \phi'_{1,2} \right),
\end{aligned}
\tag{2.5}
$$

where $v_{1,2}$ and $v'_{1,2}$ are the Vacuum Expectation Values (VEVs) of the Higgs fields $H_{1,2}$ and $\chi_{1,2}$, respectively, which radiatively break the $(B - L)$ symmetry. The real and imaginary components in (2.5) represent the CP-even (scalar) and CP-odd (pseudoscalar) Higgs states. The CP-odd neutral Higgs mass-squared matrix at the tree-level in the basis $(\phi_1, \phi_2, \phi'_1, \phi'_2)$ is given by

$$
A^2 = \begin{pmatrix}
B_\mu \tan \beta & B_\mu & 0 & 0 \\
B_\mu & B_\mu \cot \beta & 0 & 0 \\
0 & 0 & B_{\mu'} \tan \beta' & B_{\mu'} \\
0 & 0 & B_{\mu'} & B_{\mu'} \cot \beta'
\end{pmatrix},
\tag{2.6}
$$

with

$$
\begin{aligned}
B_\mu =& -\frac{1}{8} \left\{ -2 \tilde{g} g_{BL} v'^2 \cos 2\beta' + 4 M_{H_1}^2 - 4 M_{H_2}^2 \right. \\
& \left. + \left( g_1^2 + \tilde{g}^2 + g_2^2 \right) v^2 \cos 2\beta \right\} \tan 2\beta, \\
B_{\mu'} =& -\frac{1}{4} \left( -2 g_{BL}^2 v'^2 \cos 2\beta' + 2 M_{\chi_1}^2 - 2 M_{\chi_2}^2 \right. \\
& \left. + \tilde{g} g_{BL} v^2 \cos 2\beta \right) \tan 2\beta',
\end{aligned}
$$

where $\tan \beta = \frac{v_2}{v_1}$ and $\tan \beta' = \frac{v'_2}{v'_1}$. Here, $g_{BL}$ is the gauge coupling constant of $U(1)_{B-L}$ and $\tilde{g}$ is the gauge coupling constant of the mixing between $U(1)_Y$ and $U(1)_{B-L}$. Finally, $g_1$ and $g_2$ are the $U(1)_Y$ and $SU(2)_I$ gauge coupling constants, respectively.

The CP-even neutral Higgs mass-squared matrix at the tree level in the basis $(\sigma_1, \sigma_2, \sigma_1', \sigma_2')$ is given by

$$\mathcal{M}^2 = \begin{pmatrix} \mathcal{M}^2_{hH} & \mathcal{M}^2_{hh'} \\ (\mathcal{M}^2_{hh'})^T & \mathcal{M}^2_{h'H'} \end{pmatrix}, \tag{2.7}$$

where $\mathcal{M}_{hH}$ is the MSSM CP-even mass matrix which results into an SM-like Higgs boson $h$ with a mass $m_h \sim 125$ GeV and a heavier Higgs boson $H$ with a mass $m_H \sim \mathcal{O}(1 \text{ TeV})$. The additional $(B-L)$SSM mass matrix $\mathcal{M}_{h'H'}$ reads

$$\mathcal{M}^2_{h'H'} = \begin{pmatrix} m^2_{A'}c^2_{\beta'} + g^2_{BL}v_1'^2 & -\frac{1}{2}m^2_{A'}s_{2\beta'} - g^2_{BL}v_1'v_2' \\ -\frac{1}{2}m^2_{A'}s_{2\beta'} - g^2_{BL}v_1'v_2' & m^2_{A'}s^2_{\beta'} + g^2_{BL}v_2'^2 \end{pmatrix}, \tag{2.8}$$

with $c_x = \cos x$ and $s_x = \sin x$. Thus, the eigenvalues of this matrix can be given as

$$m^2_{h',H'} = \frac{1}{2}\left\{ m^2_{A'} + m^2_{Z'} \mp \sqrt{(m^2_{A'} + m^2_{Z'})^2 - 4m^2_{A'}m^2_{Z'}\cos^2 2\beta'} \right\}. \tag{2.9}$$

The mass of $h'$ can be estimated by

$$m_{h'} \simeq \left( \frac{m^2_{A'}M^2_{Z'}\cos^2 2\beta'}{m^2_{A'} + M^2_{Z'}} \right)^{\frac{1}{2}} \simeq \mathcal{O}(100 \text{ GeV}), \tag{2.10}$$

demonstrating the viability of generating a light Higgs state within the model. Finally, the matrix $\mathcal{M}_{hh'}$ can be written as

$$\mathcal{M}^2_{hh'} = \frac{1}{2}\widetilde{g}g_{BL}\begin{pmatrix} v_1v_1' & -v_1v_2' \\ -v_2v_1' & v_2v_2' \end{pmatrix}, \tag{2.11}$$

generating mixing between $(B-L)$SSM Higgs bosons and MSSM-like Higgs states. The CP-even physical Higgs mass states can be obtained by diagonalising the Higgs mass-squared matrix given by eq. (2.7) with a unitary matrix $\mathcal{R}$ as follows:

$$\mathcal{R}\mathcal{M}^2\mathcal{R}^\dagger = \text{diag}\left\{ m^2_h, m^2_{h'}, m^2_H, m^2_{H'} \right\}. \tag{2.12}$$

We then perform a parameter search, to fit the two lighter Higgs states in (2.12), as solutions consistent with the experimental reports in eqs. (2.1)-(2.3).

## 2.3 $(B-L)$SSM Predictions and HEP Software

A phenomenological analysis is typically made through a series of HEP software packages sequentially stacked, henceforth called HEP-Stack denoted $\mathcal{H}_{\text{Model}}$. Here we use SARAH [38, 39], a Mathematica package for Supersymmetric and non-Supersymmetric model building, and for each model we calculate the mass and coupling spectrum using SPheno (SP) [40, 41]. Then, MadGraph (MG) [42] is used for the computations of the cross sections relevant for the signal strength-modifiers defined in eqs. (2.1), (2.2) and (2.3).

We also use HiggsBounds (HB) [43] and HiggsSignals (HS) [44] for experimental testing of the model configurations. HB compares existing exclusion limits from Higgs searches with the model predictions of the Higgs sector, generating an upper limit to a corresponding signal cross section prediction. Therefore, with HB we can check whether a given model, whose spectrum is evaluated at a particular configuration, is excluded at the 95% Confidence Level (C.L.) by existing Higgs boson searches. This information is given by the quantity $k_0^{\text{HB}}$, with $k_0^{\text{HB}} \leq 1$ if the model configuration is accepted. HS tests, in contrast, the model prediction of a Higgs sector with an arbitrary number of Higgs bosons against the properties of the observed state as measured by the LHC experiments

ATLAS [2] and CMS [1] in 2012. The main results from HS are reported in the form of a $\chi^2_{\mathrm{HS}}$ value and the number of observables considered.

As mentioned in sections 2.1 and 2.2, our first real case study is to use our search algorithm in the $(B-L)$SSM model to allocate the excesses reported in neutral scalar searches for a 95 GeV resonance. The dimensionality of the search space is reduced by fixing $m_{Z'} = 2500$ GeV, $\tan \beta' = 1.15$, $g_{BL} = 0.53$, $g'_{BL} = 0.14$ and $\tan' \beta \leq 1.2$ [31, 33, 45], restricting the search space $\mathcal{X}$ to eight model parameters

$$\mathcal{X} = \left\{ x \in \mathbb{R}^8 : x = \left( M_0, M_{1/2}, \tan \beta, A_0, \mu, \mu', B_\mu, B_{\mu'} \right) \right\} \tag{2.13}$$

in the ranges described in Table 1.

| Parameter | Range |
|:---------:|:-----:|
| $M_0$ | $100 - 1000$ GeV |
| $M_{1/2}$ | $1000 - 4500$ GeV |
| $\tan \beta$ | $1 - 60$ |
| $A_0$ | $1000 - 4000$ GeV |
| $\mu$ | $1000 - 4000$ |
| $\mu'$ | $1000 - 4000$ |
| $B_\mu$ | $10^5 - 10^7$ |
| $B_{\mu'}$ | $10^5 - 10^7$ |

**Table 1**: Ranges defining the search space for each parameter in the $(B-L)$SSM.

We define the objective space $\mathcal{Y}$ as the space of physical observables and informative outputs generated by the HEP-Stack $\mathcal{H}_{(B-L)\mathrm{SSM}}$. These outputs are the desired targets that we seek to constrain to specific values. Specifically, we aim for the masses of the lighter Higgs particles in the model, denoted as $m_h$ and $m_{h'}$ in eq. (2.12), to have mass values of 125 GeV and 95 GeV respectively, with a certain precision. Additionally, the signal strength modifier $\mu^{\gamma\gamma}$ should satisfy the experimental value defined in (2.1). Lastly, we require the experimental checks from HB and HS to yield positive outcomes, ensuring that $k_0^{\mathrm{HB}} \leq 1$ and $\chi^2_{\mathrm{HS}} \leq 136.6$, these are the default values of the two programs. Thus, we formulate a five-dimensional objective space $\mathcal{Y}$ as follows:

$$\mathcal{Y} = \left\{ y \in \mathbb{R}^5 : y = \left( m_{h'}, m_{h^{\mathrm{SM}}}, \mu^{\gamma\gamma}, \chi^2_{\mathrm{HS}}, k_0^{\mathrm{HB}} \right) \right\} \tag{2.14}$$

with constraints defined in a vector $\tau$ for latter reference,

$$\boldsymbol{\tau}_{\gamma\gamma} = \begin{cases} m_{h^{\mathrm{SM}}} & = 125 \pm \delta m \text{ GeV} \\ m_{h'} & = 95 \pm \delta m \text{ GeV} \\ \mu^{\gamma\gamma} & = 0.27^{+0.10}_{-0.09} \\ \chi^2_{\mathrm{HS}} & \leq 136.6 \\ k_0^{\mathrm{HB}} & \leq 1 \end{cases} \tag{2.15}$$

where we take $\delta m = 5$ GeV as an acceptance window for the masses. As previously mentioned, for a particular parameter space configuration $\mathbf{x} \in \mathcal{X}$, the HEP-Stack $\mathcal{H}_{(B-L)\mathrm{SSM}}$ to evaluate is formed by SP, HB, HS and MG. High-precision spectrum calculations typically require approximately 120 seconds on average for each query to the HEP-Stack. However, in certain parameter space configurations, this time can extend up to 300 seconds. The computational cost emphasises the need for the search algorithm to prioritise *sampling efficiency*, which means maximising the number of positive parameter space configurations per $\mathcal{H}_{(B-L)\mathrm{SSM}}$ evaluations.

# 3 Active Search Formulation

The execution time of the HEP-Stack, $\mathcal{H}_{(B-L)\mathrm{SSM}}$ poses significant challenges for conventional parameter exploration methodologies such as MCMC methods. In response, parallel MCMC methods have been developed [17] or alternatives to MCMC-MH have been employed such as nested sampling [46]. These techniques struggle to identify a large and diverse set of parameter configurations that concurrently satisfy numerous constraints when limited to a small budget of calls to $\mathcal{H}_{(B-L)\mathrm{SSM}}$. To address this issue, we have articulated the problem within the framework of Active Search (AS) [47]. AS is a search methodology that utilises existing knowledge – a series of evaluations – of an objective function to identify points to sample that belong to a rare category. The rare category in this paper refers to the subset of all available parameter values $\mathbf{x}$ whose corresponding observables $\mathbf{y}$ returned by a HEP-Stack, $\mathcal{H}(\mathbf{x})$ satisfy a set of constraints denoted by $\boldsymbol{\tau}$. Here, eq. (2.15) exemplifies these constraints for $\mathcal{H}_{(B-L)\mathrm{SSM}}$. We adopt a two-stage iterative strategy. In the first stage, by iteration step $t$ we construct the dataset $\mathcal{D}_t := (\mathbf{X}_t, \mathbf{Y}_t) := (\{\mathbf{x}_j\}_{j=1}^t, \{\mathbf{y}_j\}_{j=1}^t)$. We fit a surrogate function $f : \mathbb{R}^n \to \mathbb{R}^m$, where $n$ is the dimension of the search space and $m$ the number of constraints. This function aims to approximate $\mathbf{y} \approx f(\mathbf{x})$ based on the dataset $\mathcal{D}_t$. The surrogate $f$ suggests points to sample, thus reducing the search space to query $\mathcal{H}$ and is well defined across the entire parameter space $\mathcal{X}$. In the second stage of the iteration $t$, the surrogate function will be queried by a *search policy* to create a batch of configurations $\mathbf{X}^\star \in \mathcal{X}$ that are likely to belong to the satisfactory set $\mathcal{S}$ of configurations that satisfy the set of constraints $\boldsymbol{\tau}$:

$$\mathcal{S} = \{\mathbf{x} \mid \mathbf{y} = \mathcal{H}(\mathbf{x}) \wedge y_i \succeq \tau_i, i = 1, \ldots, m\}. \tag{3.1}$$

Each sample point in $\mathbf{X}^*$ is then evaluated in the HEP-Stack $\mathcal{H}_{\mathrm{Model}}$ under study and the dataset $\mathcal{D}_{t+1}$ is updated according to $\mathcal{D}_{t+1} = \mathcal{D}_t \cup (\mathbf{X}^*, \mathbf{Y}^*)$.

The *search policy* is a crucial component in the AS framework. At each iteration $t$, the policy utilises the surrogate model $f$, fitted on $\mathcal{D}_t$, to direct exploration by selecting data points either from uncertain of underrepresented areas of the data in the search space, or to direct exploitation by choosing data points expected to yield the most useful information according to the predictions of the surrogate model. This strategic balance between the two sampling behaviors, is known as the *exploration-exploitation trade-off*.

## 3.1 Search Policy

### 3.1.1 Constraint Active Search

We adopt the conceptual and methodological advances of Constraint AS (CAS), a method developed for constrained multi-objective cases [27]. In particular, we employ the Expected Coverage Improvement (ECI) policy developed for the CAS method [27]. ECI provides a diversity measure in the search space $\mathcal{X}$ by defining a hyper-sphere of radius $r$ around a parameter space point $\mathbf{x}$, called the neighbourhood, given by

$$\mathbb{N}_r(\mathbf{x}) = \{\mathbf{x}' : d(\mathbf{x}, \mathbf{x}') < r\} \tag{3.2}$$

where $d$ is the Euclidean distance. The total neighbourhood for a set of input points $\mathbf{X}$ given a dataset $\mathcal{D}$ is defined as the coverage neighbourhood,

$$\mathbb{N}_r(\mathbf{X}) = \bigcup_{\mathbf{x} \in \mathbf{X}} \mathbb{N}_r(\mathbf{x}),$$

Thus, the volume utility function can be defined,

$$u_{\mathcal{S}}(\mathcal{D}) = \mathrm{Vol}(\mathbb{N}_r(\mathcal{D}) \cap \mathcal{S}) \tag{3.3}$$

where $u_{\mathcal{S}}$ measures the total volume of $S$ covered by the neighborhood $\mathbb{N}_r(\mathcal{D})$. We also define the total volume covered as $u_{\mathrm{T}}$. CAS aims to discover the dataset $\mathcal{D}$ that covers as much volume of the satisfactory region $\mathcal{S}$ as possible through the maximisation of the ECI policy function,

$$\alpha\left(\mathbf{x} \mid \mathcal{D}\right) = \mathbb{E}_{\mathbf{y}}\left[u_{\mathcal{S}}\left(\mathcal{D}_t \cup (\mathbf{x}, \mathbf{y})\right) - u_{\mathcal{S}}\left(\mathcal{D}_t\right)\right] \tag{3.4}$$

Therefore, at time step $t$ of the search, the policy proposes a configuration $\mathbf{x}^*$ through

$$\mathbf{x}^* = \arg\max_{\mathbf{x} \in \mathcal{X}} \alpha\left(\mathbf{x} \mid \mathcal{D}_t\right) \tag{3.5}$$

### 3.1.2 Batch Evaluation

Originally, the optimisation of ECI is made point-wise and sequentially, with well established routines, such as L-BFGS-B [48]. These classical optimisation methods are sufficient, given the original focus of ECI on experiment design, where the global search budget was relatively low, with order of $\mathcal{O}(10^2)$ points. However, in this work, our goal is to densely populate $\mathcal{S}$, *i.e.*, to collect as many samples from $\mathcal{S}$ as possible. Since ECI, eq. (3.4), depends directly on the size of the dataset $\mathcal{D}_t$ at iteration $t$ of the search process, the time required to optimise it increases accordingly. To execute eq. (3.5) efficiently we use the Tree-structured Parzen Estimator (TPE) algorithm [49]. TPE is a variant of BO, commonly used for hyper-parameter optimisation in Machine Learning.

The TPE optimisation process evaluates ECI a number of times to generate a historical dataset $\tilde{\mathcal{D}}_{\mathrm{TPE}} := (\tilde{\mathbf{X}}, \alpha(\tilde{\mathbf{X}}))$ known as trials, from which the optimal parameter value $\mathbf{x}^*$ in eq. (3.5) is identified. $\tilde{\mathcal{D}}_{\mathrm{TPE}}$ contains parameter configurations with sub-optimal ECI values, but which lie within the satisfactory region $\mathcal{S}$. Hence, evaluating this subset of sub-optimal configurations on $\mathcal{H}$ accelerates the collection of parameter space points within $\mathcal{S}$. For this purpose, instead of selecting a single estimated maximal point $\mathbf{x}^*$ to evaluate $\mathcal{H}(\mathbf{x}^*)$, we sample a set of $N_{\mathrm{batch}}$ parameter points $\mathbf{X}^* = [\mathbf{x}_0^*, \mathbf{x}_1^*, .., \mathbf{x}_{N_{\mathrm{batch}}}^*]$ from $\tilde{\mathcal{D}}_{\mathrm{TPE}}$ and evaluate every point on $\mathcal{H}$ in each iteration of the search. This method, referred to as batch evaluation, accelerates the filling of the $\mathcal{S}$ region in the search process, as the HEP-Stack $\mathcal{H}$ allows parallel evaluations of each configuration in the batch.

The batch $\mathbf{X}^*$ is sampled according to a rank-based sampling strategy that interpolates between pure greedy prioritisation and uniform random sampling, initially developed in the context of Reinforcement Learning [50] and called stochastic prioritisation. In this scheme, each $\tilde{\mathbf{x}} \in \tilde{\mathbf{X}}$ is assigned a rank $rk(\tilde{\mathbf{x}})$ so that,

$$rk(\tilde{\mathbf{x}}_i) \leq rk(\tilde{\mathbf{x}}_j) \text{ for } \alpha(\tilde{\mathbf{x}}_i) \geq \alpha(\tilde{\mathbf{x}}_j). \tag{3.6}$$

This determines the probability of sampling

$$P(\tilde{\mathbf{x}}_i) = \frac{rk(\tilde{\mathbf{x}}_i)^{-\beta}}{\sum_{\tilde{\mathbf{x}}_j \in \tilde{\mathbf{X}}} rk(\tilde{\mathbf{x}}_j)^{-\beta}}, \tag{3.7}$$

where $\beta$ determines the extent to which $\alpha(\tilde{\mathbf{x}})$ prioritises selection, with $\beta = 0$ corresponding to uniform sampling. Thus, points with higher ECI value will be more likely to be sampled, while also enabling exploration of the parameter space by sampling low value ECI.

### 3.2 Surrogate Models

We utilise an independent Gaussian Process (GP) as a surrogate model for each objective function. For $\mathcal{H}_{(B-L)\mathrm{SSM}}$, this approach involves approximating each of the five objective variables in $\mathcal{Y}$ as defined in eq. (2.14). Using GPs is a common practice in BO [51] approaches. GPs model the entire distribution of possible functions that can describe a given set of observations as a multivariate Gaussian distribution [29, 52]. This provides not only a point estimate of an objective but also

quantifies the uncertainty associated with that estimate. This uncertainty is crucial for search methods, as it provides key information that guides the control over the *exploitation-exploration trade-off*, as described in section 3. A GP defines a probability distribution over functions $f(\mathbf{x})$ which is specified completely by the mean function $\mu(\mathbf{x})$ and covariance function $k(\mathbf{x}, \mathbf{x}')$ and can be written as,

$$f(\mathbf{x}) \sim \mathcal{GP}(\mu(\mathbf{x}), k(\mathbf{x}, \mathbf{x}')) \tag{3.8}$$

with,

$$\mu(\mathbf{x}) = \mathbb{E}[f(\mathbf{x})]$$
$$k(\mathbf{x}, \mathbf{x}') = \mathbb{E}[(f(\mathbf{x}) - \mu(\mathbf{x}))(f(\mathbf{x}') - \mu(\mathbf{x}'))]$$

where $\mathbf{x}$ are values in the input domain (here, parameters in Table 1) and $(\mathbf{x}, \mathbf{x}')$ all possible pairs of parameter values. The covariance function, also known as the kernel function, specifies how the output of the function at one input $\mathbf{x}$ covaries with the output at another input $\mathbf{x}'$. The choice of kernel function determines the smoothness, periodicity and other structural properties of the functions sampled from the GP prior family. In this work we use the Matérn kernel class of covariance functions, defined as

$$k_\ell^\nu(x, x') = \frac{2^{1-\nu}}{\Gamma(\nu)} \left( \frac{\sqrt{2\nu} |x - x'|}{\ell} \right)^\nu K_\nu \left( \frac{\sqrt{2\nu} |x - x'|}{\ell} \right), \tag{3.9}$$

where $\ell$ is a parameter that controls the length-scale over which correlations persist whereas $\nu$ parameter controls the level of smoothness of the modified Bessel function $K_\nu$.

For any $N$ observed points $\{x_i\}_{i=1}^N$ $K(x_i, x_j)$ defines the matrix elements of the covariance matrix of a $N$-dimensional Gaussian, where $K$ is the Matérn kernel $k_\ell^\nu$. For an observed data-set $\mathcal{D} = \{X, Y\}$ of parameter values $X$ and target evaluations $Y$, the targets $Y^*$ corresponding to parameter values $X^*$ are described by a posterior predictive distribution using Bayesian inference. Considering the case where the observations $Y$ are noise free and the prior mean is zero, the joint distribution of the training outputs $Y$ and the test outputs $Y^*$ is also a Gaussian with covariance matrix

$$\begin{bmatrix} Y \\ Y^* \end{bmatrix} \sim \mathcal{N} \left( \begin{bmatrix} 0 \\ 0 \end{bmatrix}, \begin{bmatrix} K(X, X) & K(X^*, X) \\ K(X, X^*) & K(X^*, X^*) \end{bmatrix} \right) \tag{3.10}$$

where $K(X, X)$ defines the sub-matrix representing the covariance corresponding to $\mathcal{D}$ obtained by evaluating the kernel function $k_\ell^\nu$ on the pairwise combinations of outputs for each element in $X$. $K(X^*, X)$ and $K(X^*, X^*)$ represents elements of the covariance between training points $X$ and unseen points $X^*$. The posterior predictive distribution of $Y^*$ for the test points $X^*$ is conditioned on the data-set $\mathcal{D}$:

$$p(Y^* \mid X^*, X, Y) = \frac{p(Y^*, Y \mid X, X^*)}{p(Y \mid X, X^*)}. \tag{3.11}$$

By inverting the block covariance matrix, we obtain the mean vector and covariance matrix of the distribution over predictions $Y^*$

$$\begin{aligned} \mu(Y^* \mid X^*, \mathcal{D}) &= K(X^*, X) K(X, X)^{-1} Y \quad \text{and} \\ \Sigma(Y^* \mid X^*, \mathcal{D}) &= K(X^*, X^*) - K(X^*, X) K(X, X)^{-1} K(X, X^*), \end{aligned} \tag{3.12}$$

respectively. The optimal set of hyper-parameters $\{\ell, \nu\}$ for the Matérn kernel function $k_\ell^\nu$ is determined by maximising the log marginal likelihood, expressed as:

$$\log p(Y \mid X, \boldsymbol{\theta}) = -\frac{1}{2} Y^\top k_\ell^\nu(X, X)^{-1} Y - \frac{1}{2} \log |k_\ell^\nu(X, X)| - \frac{n}{2} \log 2\pi \tag{3.13}$$

where $n$ is the number of data points. Consequently, the training phase of the GPs involves maximising 3.13. In our study, we perform a training phase for the GPs in each iteration of the search process.

### 3.3 b-CASTOR

Building on the components described in the previous sections, we now introduce our algorithm, b-CASTOR, which stands for **b**atched **C**onstrained **A**ctive **S**earch with **T**PE **O**ptimisation and **R**ank based sampling.

The algorithm starts by initialising a specific number of points[5], denoted as $N_0$, creating the initial dataset $\mathcal{D}_0$. Each search iteration involves fitting an independent Gaussian Process (GP) model to each objective, using the current observation dataset $\mathcal{D}_i$. The ECI policy (3.4) is then optimised by the TPE algorithm over policy evaluations $\tilde{\mathcal{D}}_{\mathrm{TPE}}$ generated over $N_{\mathrm{TPE}}$ trials. These evaluations are ordered by their ECI values and a priority is assigned to each point in this set as described in section 3.1.2. A batch of points $\mathbf{X}^*$ is selected from $\tilde{\mathcal{D}}_{\mathrm{TPE}}$ based on the probability distribution constructed with the priorities. Each sample point in $\mathbf{X}^*$ is then evaluated in the HEP-Stack $\mathcal{H}_{\mathrm{Model}}$ under study and the dataset $\mathcal{D}_{i+1}$ is updated according to $\mathcal{D}_{i+1} = \mathcal{D}_i \cup (\mathbf{X}^*, \mathbf{Y}^*)$. This iterative process continues until it reaches the pre-established number of total iterations, $T_{\mathrm{iter}}$, or when a total number of samples is met, denoted by $T_{\mathrm{samples}}$. The pseudo-code is described in Algorithm 1.

---

**Algorithm 1** b-CASTOR

---

1: Initialise parameters: $N_0$ (number of initial points), $N_{\mathrm{TPE}}$ (number of TPE trials), $N_{\mathrm{batch}}$ (batch size), $T$ (number of search iterations) and $\beta$ (prioritisation parameter).
2: Generate a initial dataset $\mathcal{D}_0$ with $N_0$ points.
3: **for** $i = 1, T$ **do**
4:     Fit Surrogate models to $\mathcal{D}_i$.
5:     Optimise ECI using TPE algorithm with $N_{\mathrm{TPE}}$ trials generating $\tilde{\mathcal{D}}_{\mathrm{TPE}}$.
6:     Assign a rank $rk(\tilde{\mathbf{x}})$ for each $\tilde{\mathbf{x}} \in \tilde{\mathbf{X}}$ in $\tilde{\mathcal{D}}_{\mathrm{TPE}}$ following eq. (3.6).
7:     Sample a batch $\mathbf{X}^* = [\mathbf{x}_0^*, \mathbf{x}_1^*, .., \mathbf{x}_{N_{\mathrm{batch}}}^*]$ from $\tilde{\mathcal{D}}_{\mathrm{TPE}}$ using probabilities $P(\tilde{\mathbf{x}}_i)$ from eq. (3.7).
8:     Evaluate $\mathbf{X}^*$ in HEP-Stack $\mathcal{H}_{\mathrm{Model}}$
9:     Update $\mathcal{D}_{i+1} = \mathcal{D}_i \cup (\mathbf{X}^*, \mathbf{Y}^*)$
10: **end for**

---

In section 4, we perform a grid hyper-parameter search for $N_{\mathrm{TPE}}$, $\beta$ and $r$ (defined in eq. (3.2)) for a test objective function. We also implement a linear decay in $r = \{r_{\mathrm{initial}}, r_{\mathrm{final}}\}$ along the search process. This configuration on $r$ enables an early discovery of the $\mathcal{S}$ region and subsequent fine resolution filling. The values of $r_{\mathrm{initial}}$ and $r_{\mathrm{final}}$ are also considered in the grid hyper-parameter search.

### 3.4 Performance Study

We compare our algorithm with a Markov chain Monte Carlo (MCMC) method, specifically, the Metropolis-Hastings (MH) algorithm [19, 30] (hereafter, denoted my MCMC-MH). The sampling with the MH is performed with the construction of a joint likelihood for the objectives. For each objective $y_i$, constrained by either a threshold $a$ or a window with limits $[a, b]$, we define a likelihood given by

$$\mathcal{L}(y_i) = \begin{cases} \sigma(y_i, a) & y_i > a \\ 1 - \sigma(y_i, a) & y_i < a \\ \sigma(y_i, a) - \sigma(y_i, b) & a < y_i < b \end{cases} \tag{3.14}$$

where $\sigma$ is the Sigmoid function and is defined as,

$$\sigma(y, a) = \frac{1}{1 + e^{-(y-a)/\epsilon}} \tag{3.15}$$

---

[5]We use a Sobol sequence [53], a quasi-random low-discrepancy sequence of points in the search space.

Here $\epsilon$ is a parameter that controls the smoothness of the Sigmoid function and $a$ shifts the center of the Sigmoid. Then, the total likelihood for a point $(\mathbf{x}, \mathbf{y})$ used for MCMC-MH sampling is defined as,

$$\mathcal{L}(\mathbf{y}) = \prod_i \mathcal{L}(y_i) \tag{3.16}$$

Given an initial proposal step-size, we adjust this step-size actively to reach a target acceptance rate of 0.234 [54]. The scale is increased by 10% if the acceptance rate is above a the threshold and decreased by 10% if the acceptance rate is below.

Lastly, two main metrics are monitored for both search strategies. The satisfactory points per objective function call and the ratio of satisfactory points to total points in the dataset per call.

## 3.5  Technical Implementation

We employ the Expected Coverage Improvement acquisition function implementation provided by the BoTorch library [55]. BoTorch provides various components required in BO, including acquisition function optimisation and optimisation metrics. Additionally, it incorporates probabilistic models from the library GPytorch [56] a library for scalable GP inference, built on PyTorch [57]. We utilise the TPE implementation available in Optuna [58], an open source hyperparameter optimisation framework. The code developed for this work has been incorporated into a general library for AS in order to perform phenomenological studies and will be publicly released alongside an upcoming manual for it.

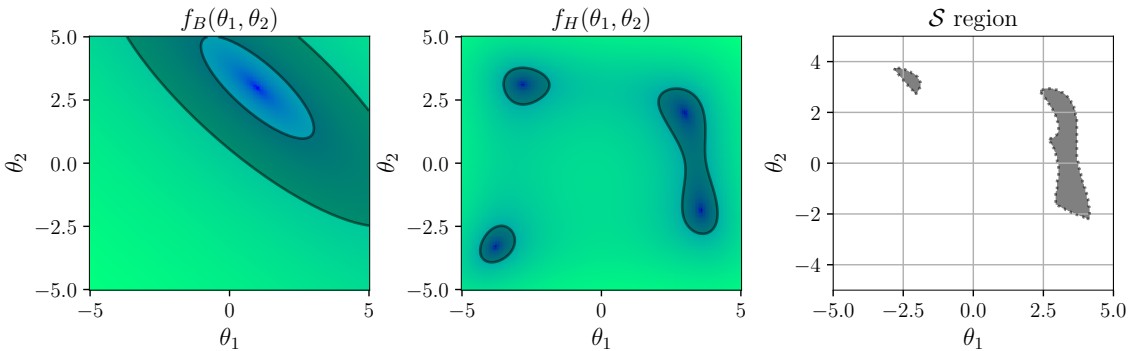

**Figure 1**: Ground truth for the 2D double objective test function $\mathbf{f}_{BH}(\theta_1, \theta_2)$ as per eq. (4.1), featuring contour levels to demonstrate the constraints on the objectives. On the left, the $\mathcal{S}$ region within the search space is depicted.

## 4  Results

### 4.1  Double-objective 2D Test Function

We constructed a simple double-objective, 2D test function, denoted $\mathbf{f}_{BH}(\theta)$, from the Booth and Himmelblau functions [59], a uni- and multi-modal function, respectively, defined as

$$\mathbf{f}_{BH}(\theta) = \begin{cases} f_B(\theta_1, \theta_2) = \log\left[(\theta_1 + 2\theta_2 - 7)^2 + (2\theta_1 + \theta_2 - 5)^2\right], \\ f_H(\theta_1, \theta_2) = \log\left[\left(\theta_1^2 + \theta_2 - 11\right)^2 + \left(\theta_1 + \theta_2^2 - 7\right)^2\right]. \end{cases} \tag{4.1}$$

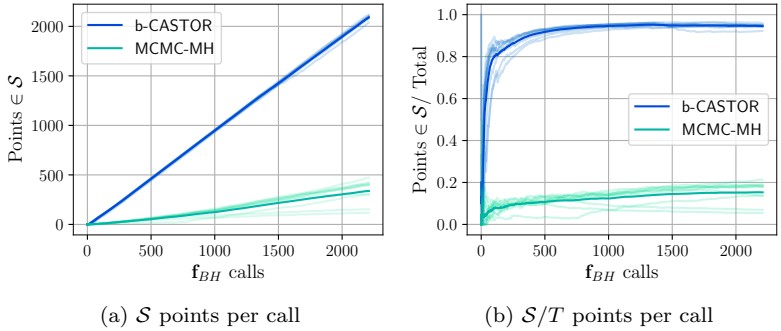

(a) $\mathcal{S}$ points per call    (b) $\mathcal{S}/T$ points per call

**Figure 2**: Performance metrics across ten independent runs for b-CASTOR (blue) and MCMC-MH (green), with mean values depicted in darker shades.

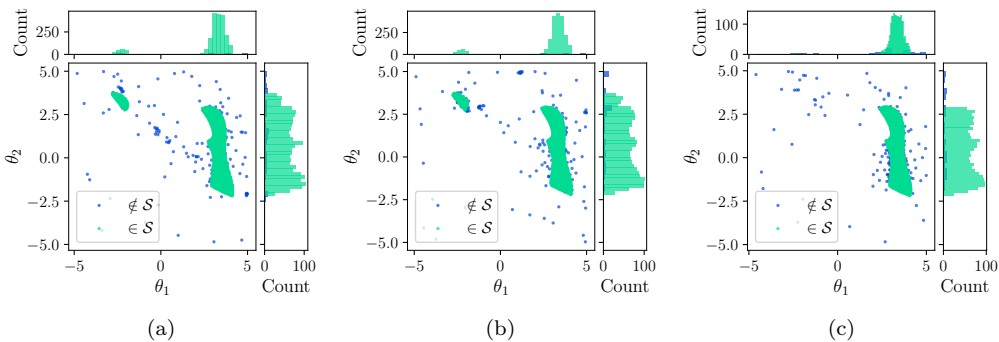

(a)    (b)    (c)

**Figure 3**: b-CASTOR results for different independent runs.

The parameter space is defined as $\theta \in [-5, 5]$. Additionally, the search process is subject to the constraints on the objectives given by

$$\boldsymbol{\tau}_{\mathbf{f}_{BH}} = \begin{cases} f_B(\theta_1, \theta_2) & = 2 \pm 1, \\ f_H(\theta_1, \theta_2) & < 3. \end{cases} \tag{4.2}$$

This simple set-up imitates the complexity of sparse and disconnected satisfactory regions in the search space of multiple objectives. The ground truth satisfactory regions for both objectives in $\mathbf{f}_{BH}(\theta_1, \theta_2)$ are shown in Figure 1.

As mentioned in section 3.3, we performed a hyper-parameter grid search for the number of policy evaluations $N_{\mathrm{TPE}}$, priority scaling $\beta$ (eq. (3.7)) as well as the parameter resolution limits $r_{\mathrm{initial}}$ and $r_{\mathrm{final}}$ in eq. (3.2). The hyper-parameter search space is defined by the set of values in Table 2, together with the fixed hyper-parameters. We selected $N_{\mathrm{TPE}} = 500$, $\beta = 2$, $r_{\mathrm{initial}} = 0.02$ and $r_{\mathrm{final}} = 0.0002$. For the MCMC-MH algorithm, an initial step size of 0.4 was established, optimised for the potential discovery of the disconnected $\mathcal{S}$ regions. In order to evaluate the consistency of convergence for both the b-CASTOR and MCMC-MH algorithms, we performed 10 searches. For this we have set the hyper-parameters as mentioned and restricted the search to 2200 calls to the objective test function. This was done in anticipation of the application to querying the HEP-Stack, where the complexity of the search is dominated by time needed for evaluating each query. The performance metrics are presented in Figure 2. b-CASTOR achieved an average of 2090 satisfactory parameter values by the end of the search (excluding the initial dataset), indicating that 94.57% of

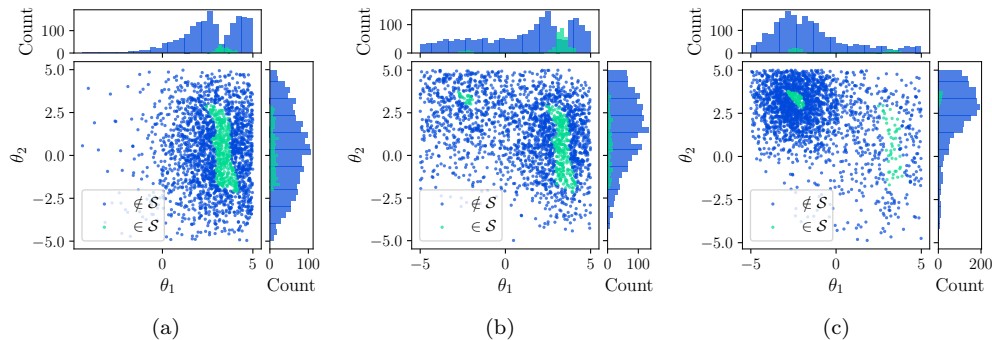

**Figure 4**: MCMC-MH results for different independent runs.

the objective function calls resulted in parameter space configurations that satisfy the constraints on the objectives. In contrast, MCMC-MH recorded an average of 338 satisfactory points at the end of the search, corresponding to 15.29% of the total calls. Out of the 10 searches conducted, the results of three runs per algorithm are illustrated in Figures 3 and 4.

| Hyper-parameter | Value |
|---|---|
| $N_0$ | 10 |
| $N_{\text{batch}}$ | 10 |
| $T_{\text{samples}}$ | 2200 |
| $N_{\text{TPE}}$ | $\{100, 300, 500\}$ |
| $\beta$ | $\{1, 2\}$ |
| $(r_{\text{initial}}, r_{\text{final}})$ | $\{(0.2, 0.02), (0.02, 0.002), (0.02, 0.0002)\}$ |

**Table 2**: b-CASTOR hyper-parameters for the search in $\mathbf{f}_{\text{BH}}$. Values in brackets define the hyper-parameter grid search space.

The superior performance of b-CASTOR over MCMC-MH is evident from the results shown in these plots. A high number of TPE trials allows the collection of a large number of samples, which are distributed across high values of the ECI acquisition function. By setting a quadratic priority parameter, $\beta = 2$, the ranked-sampling strategy is adjusted to strictly favour exploitation over exploration. Consequently, in each search iteration, the proposed batch $\mathbf{X}^*$ is more likely to contain a set of parameter values that meet all the constraints – a satisfactory set – and show sufficient variability in the parameters found – a diverse set – leading to a high *sample efficiency* of the search. Figure 3 demonstrates that with b-CASTOR the $\mathcal{S}$ region can be accurately characterised without having to explore the entire search space. In contrast, MCMC-MH in Figure 4 exhibits lower sample efficiency, with many evaluations spread across areas surrounding the $\mathcal{S}$ region.

Furthermore, as mentioned in section 3.3, setting the radius parameter $r$ in eq. (3.2) to have a linear decay allows early exploration when the initial size of the radius is relatively big compared to the search space leading to a comprehensive initial estimation of the satisfactory region. Subsequently, since the radius decreases with each iteration, the $\mathcal{S}$ region gets densely populated. However, sections of the $\mathcal{S}$ region not identified in the initial low-resolution phase are likely to remain undiscovered. This behaviour is observed in Figure 3 (c), where a small segment of the $\mathcal{S}$ region in the top-left quadrant remains undetected. Nonetheless, b-CASTOR extends its exploration to uncover additional sections of the $\mathcal{S}$ region once the previously discovered areas are fully

covered, by leveraging the uncertainty estimation of the surrogate model on unexplored regions in the search space.

### 4.2 The $(B-L)$SSM and a $95$ GeV Higgs Boson

We now examine the first phenomenology case study with $(B-L)$SSM model, searching for model configurations that can explain the $\gamma\gamma$ anomaly, defined in equation (2.1). As described in section 2.3, the objective function is defined as follows:

$$\mathcal{H}_{(B-L)\text{SSM}} : \big(M_0, M_{1/2}, \tan\beta, A_0, \mu, \mu', B_\mu, B_{\mu'}\big) \to \big(m_{h'}, m_{h^{\text{SM}}}, \mu^{\gamma\gamma}, \chi^2_{\text{HS}}, k_0^{\text{HB}}\big)$$

where each objective is constrained by $\tau$ specified in eq. (2.15). One issue that arises in sampling methods for BSM phenomenology is to guarantee the physical validity of each parameter configuration. In certain BSM models, SP fails to converge to a physical spectra for a significant portion of points within the search space, being the case for the $(B-L)$SSM. In [20] they addressed this challenge by employing a NN classifier as a preliminary step to regression on observables. In [60] they include these points as points outside the satisfactory set, assigning them a zero likelihood. In this work, for we discard the non-physical points and only work with valid parameter space configurations.

The hyper-parameters for the b-CASTOR search are outlined in table 3. We have allocated a greater number of TPE trials compared to the test function. This decision is based on findings from section 4.1, which demonstrated that an increase in TPE trials enhances the sample efficiency of the b-CASTOR search process.

Figure 5 illustrates the performance of both algorithms throughout the search process. b-CASTOR identified 1636 satisfactory configurations, constituting up to 50% of the total $\mathcal{H}_{(B-L)\text{SSM}}$ calls, which amounted to 3240. In contrast, MCMC-MH was able to find only 25 satisfactory configurations, representing a mere 0.008% of the total calls.

| Hyper-parameter | Value |
|---|---|
| $N_0$ | 400 |
| $N_{\text{batch}}$ | 30 |
| $T_S$ (Total samples) | 3240 |
| $N_{\text{TPE}}$ | 2500 |
| $\beta$ | 2 |
| $(r_{\text{initial}}, r_{\text{final}})$ | $(10^{-2}, 10^{-6})$ |

**Table 3**: b-CASTOR hyper-parameters for the search in $\mathcal{H}_{(B-L)\text{SSM}}$.

The results obtained from each algorithm, b-CASTOR and MCMC-MH, are depicted in Figures 6 and 7, respectively, using corner plots. These plots constitute a triangular grid of 2-Dimensional (2D) scatter plots for each pair of variables, supplemented by marginal histograms for individual variables. The figures integrate a selection of relevant dimensions from both the search space, $\mathcal{X}$, and the objective space, $\mathcal{Y}$; specifically, $\{M_0, M_{1/2}, \tan\beta, A_0\}$ from $\mathcal{X}$ and $\{m_{h'}, m_{h^{\text{SM}}}, \mu^{\gamma\gamma}\}$ from $\mathcal{Y}$. The dimensions of interest in the objective space are highlighted with green axes titles and green bands marking the constraints, defined in eq. (2.15), within each plot.

Upon comparing Figures 6 and 7, it is evident that, with an equivalent number of function calls, b-CASTOR comprehensively characterises the $\mathcal{S}$ region in a sample efficient manner, in contrast to MCMC-MH, which encounters difficulties in accurately characterising this region. b-CASTOR

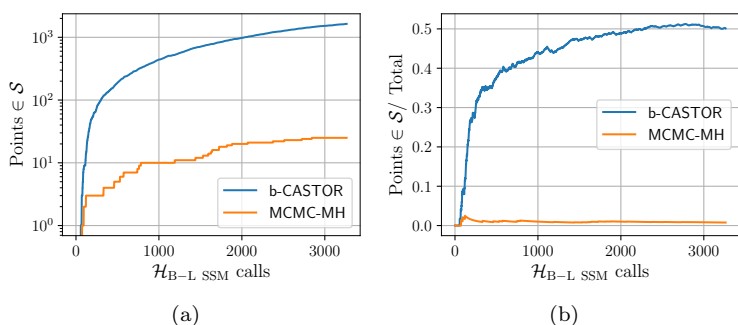

**Figure 5**: Performance metrics for b-CASTOR (blue) and MCMC-MH (orange), for the search in $\mathcal{H}_{(B-L)\text{SSM}}$ fitting $\mu_{\gamma\gamma}^{\text{exp}}$.

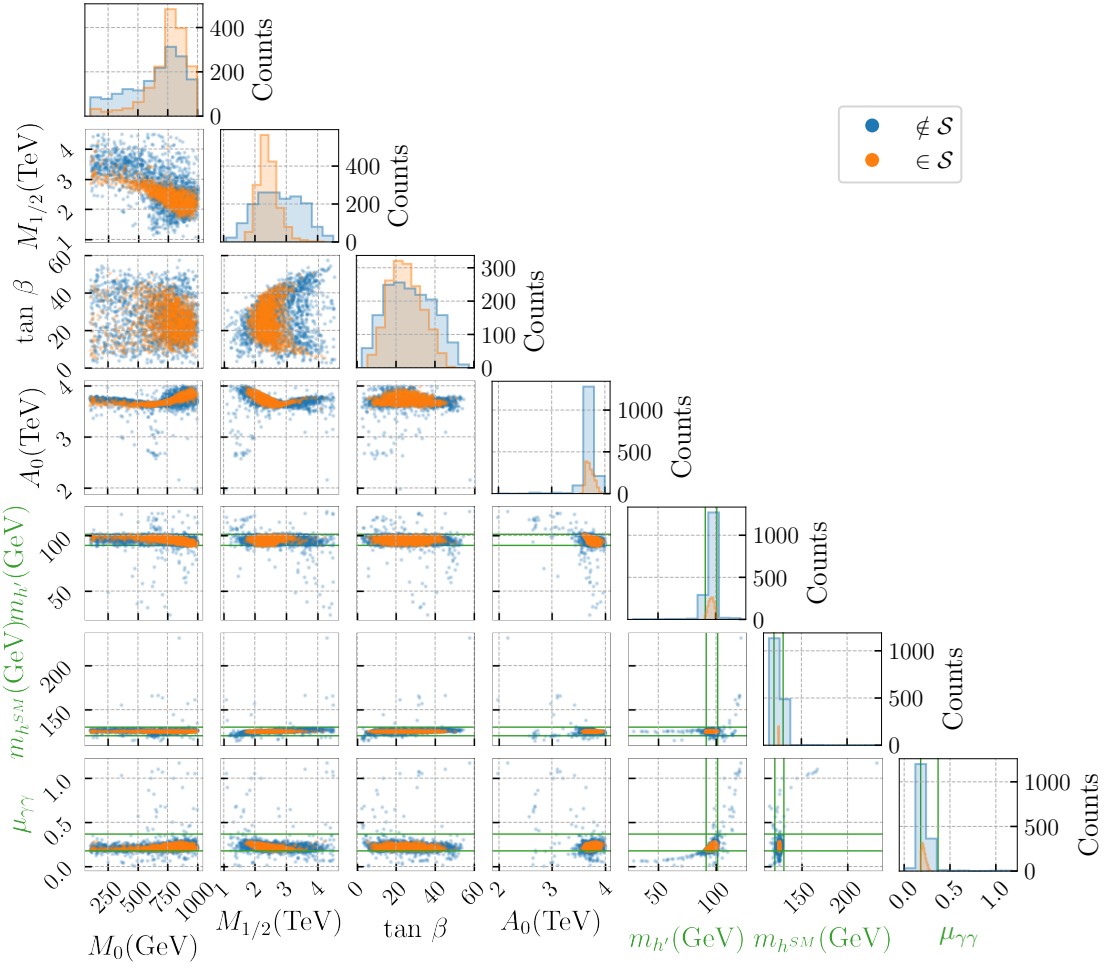

**Figure 6**: b-CASTOR results for $\mathcal{H}_{(B-L)\text{SSM}}$ fitting $\mu_{\gamma\gamma}^{\text{exp}}$. A selection of dimensions from both the search and objective spaces is presented; $\{M_0, M_{1/2}, \tan\beta, A_0\}$ (with black axis titles) and $\{m_{h'}, m_{h^{\text{SM}}}, \mu^{\gamma\gamma}\}$ (with green axis titles), respectively. Green bands represent the experimental constraints on the objectives.

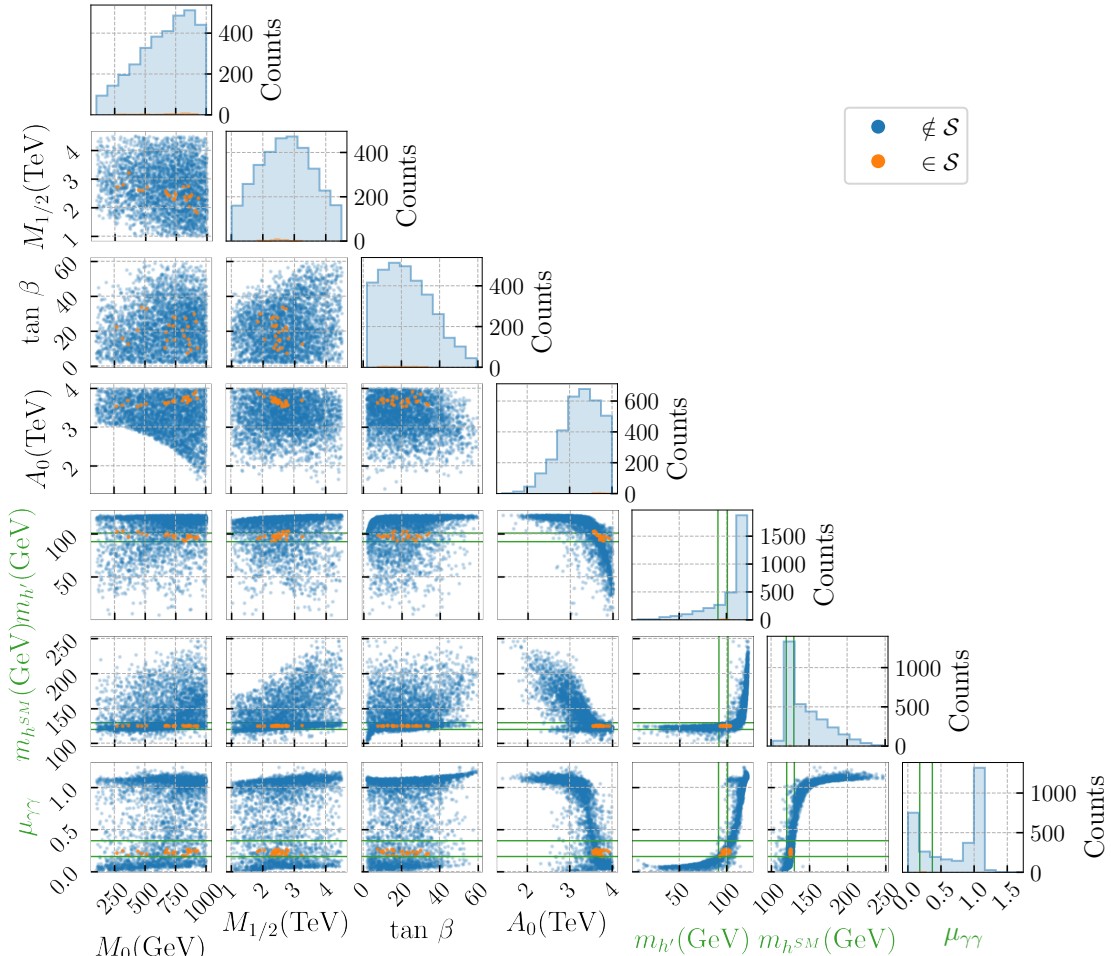

**Figure 7**: MCMC-MH results for $\mathcal{H}_{(B-L)\mathrm{SSM}}$ fitting $\mu_{\gamma\gamma}^{\mathrm{exp}}$. A selection of dimensions from both the search and objective spaces is presented; $\{M_0, M_{1/2}, \tan\beta, A_0\}$ (with black axis titles) and $\{m_{h'}, m_{h^{\mathrm{SM}}}, \mu^{\gamma\gamma}\}$ (with green axis titles), respectively. Green bands represent the experimental constraints on the objectives.

concentrates on exploring the neighbourhood areas of the identified portion of the $\mathcal{S}$ region, simultaneously ensuring these portions are densely populated. Conversely, MCMC-MH explores a more extensive area within the valid search space, centred around a few $\mathcal{S}$ points, as we can read from the blue marginal plots of the search space dimensions $\{M_0, M_{1/2}, \tan\beta, A_0\}$ in Figure 7, without prioritising samples within the $\mathcal{S}$ region. This sampling behaviour results from the nature of the Gaussian proposal distribution in the MCMC-MH algorithm.

Subsequently, $\mu_{bb}$ can be included in the objectives, using the experimental value defined in eq. (2.2). The objective space is then defined as follows:

$$\mathcal{Y}_{\gamma\gamma+bb} = \left\{ y \in \mathbb{R}^6 : y = \left( m_{h'}, m_{h^{\mathrm{SM}}}, \mu_{\gamma\gamma}, \mu_{bb}, \chi^2_{\mathrm{HS}}, k_0^{\mathrm{HB}} \right) \right\} \tag{4.3}$$

constrained to

$$\boldsymbol{\tau}_{\gamma\gamma+bb} = \begin{cases} m_{h^{\mathrm{SM}}} & = 125 \pm \delta m \text{ GeV} \\ m_{h'} & = 95 \pm \delta m \text{ GeV} \\ \mu^{\gamma\gamma} & = 0.27^{+0.10}_{-0.09} \\ \mu_{bb} & = 0.117 \pm 0.057 \\ \chi^2_{\mathrm{HS}} & \leq 136.6 \\ k_0^{\mathrm{HB}} & \leq 1 \end{cases} \tag{4.4}$$

where we considered $\delta m = 5$ GeV. Incorporating $\mu_{bb}$ necessitates the addition of the relevant cross-section, computed by MG, thereby increasing the computational cost associated with $\mathcal{H}_{(B-L)\mathrm{SSM}}$. Consequently, we conduct the b-CASTOR search utilising the same hyper-parameters as in Table 3, albeit with a reduced total sample budget of $\sim 1000$ function calls, owing to the increased computational cost. b-CASTOR achieved 5% of satisfactory points, which corresponds to $\sim 50$ points. Under identical settings, MCMC-MH failed to identify any satisfactory configurations restricted to the same number of function calls. We allowed MCMC-MH to continue running until it located at least one satisfactory point, achieving a rate of approximately 1 in 4000. The results for the discovered $\mathcal{S}$ points by b-CASTOR are shown in Figure 8.

Finally, we conducted a b-CASTOR search for the three reported signal-strength modifiers $\mu_{\gamma\gamma}$, $\mu_{bb}$ and $\mu_{\tau\tau}$, with experimental values defined in eqs. (2.1), (2.2) and (2.3), respectively. The search was unable to identify satisfactory parameter space configurations, suggesting that the $(B-L)\mathrm{SSM}$ cannot simultaneously accommodate the three signals, within the designated search parameter space $\mathcal{X}$, as defined in eq. (2.13), with parameter ranges specified in Table 1.

## 5 Conclusions

In this paper, we have introduced b-CASTOR, a novel multi-objective active search method for computationally expensive BSM scenarios. It effectively identifies $\mathcal{S}$, the satisfactory region of the corresponding parameter space that can accommodate a combination of desired values for the objectives while achieving high sample efficiency due to the use of probabilistic surrogate models for approximating the multiple objectives. It provides sample diversity in the search space by leveraging the ECI acquisition function, a volume based metric that operates to maximise the covered volume of the $\mathcal{S}$ region.

The paper evaluated b-CASTOR using two case studies. The first case involved a double-objective and a 2D test function designed to replicate the complexity of sparse and disjoint satisfactory regions. The second case focused on BSM phenomenology, specifically employing the $(B-L)\mathrm{SSM}$ scenario to allocate the observed signal at approximately 95 GeV from Higgs searches. Specifically, in this work, we attempt to find explanations to three possible data anomalies emerged at the above mass value in the $\gamma\gamma$, $\tau\tau$ (at the LHC) and $b\bar{b}$ (at LEP) invariant masses.

We conducted a comparative analysis between our proposed search method, b-CASTOR, and a MCMC-MH. Our findings illustrate the effectiveness of our algorithm in characterising the $\mathcal{S}$ region within the parameter space of the $(B-L)\mathrm{SSM}$ model, by efficiently finding solutions herein in the case of the longest established anomaly, i.e. the one in the $\gamma\gamma$ channel, obtained in the form a light scalar state, $h$. However, considering updated experimental data for the channels $h \to \tau\tau$ and $h \to b\bar{b}$, our approach did not find points capable of simultaneously explaining all three channels. Nonetheless, it identified nine points capable of satisfying the signal strengths $\mu_{\gamma\gamma}$ and $\mu_{bb}$ concurrently.

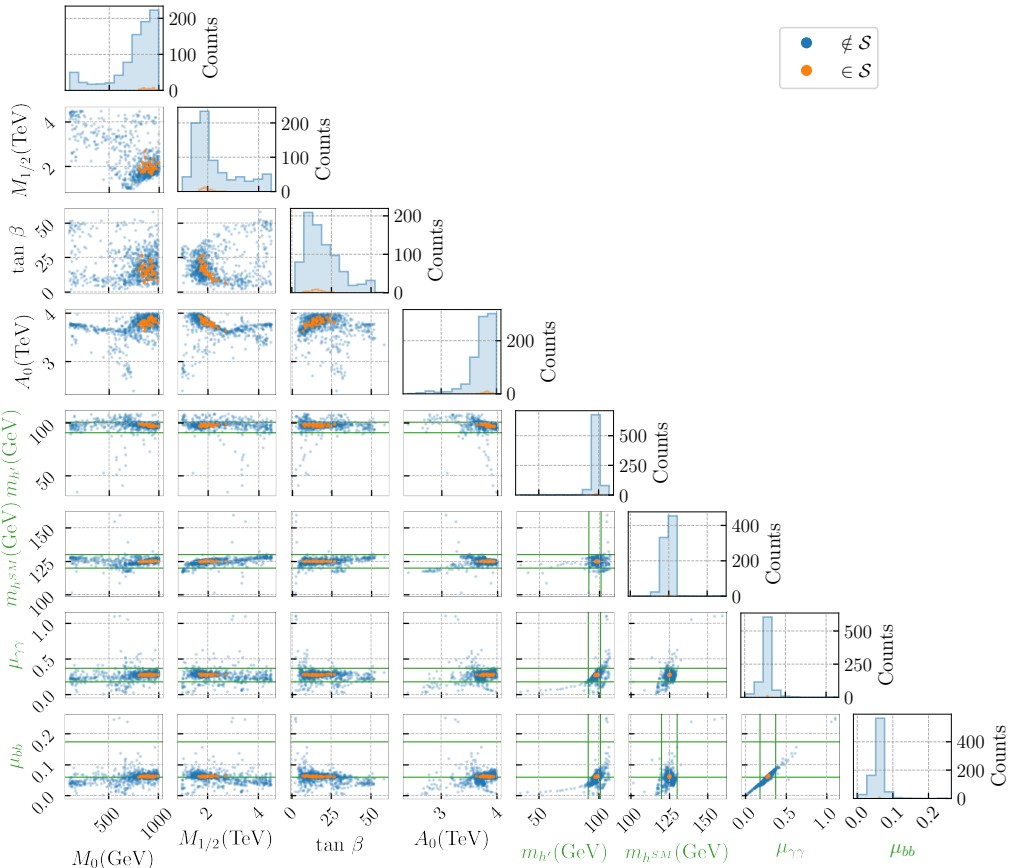

**Figure 8**: b-CASTOR results for $\mathcal{H}_{(B-L)\mathrm{SSM}}$ fitting $\mu_{\gamma\gamma}^{\mathrm{exp}}$ and $\mu_{bb}^{\mathrm{exp}}$. A selection of dimensions from both the search and objective spaces is presented; $\{M_0, M_{1/2}, \tan\beta, A_0\}$ (with black axis titles) and $\{m_{h'}, m_{h^{\mathrm{SM}}}, \mu^{\gamma\gamma}, \mu^{bb}\}$ (with green axis titles), respectively. Green bands represent the experimental constraints on the objectives.

b-CASTOR is robust against the choice of the resolution parameter $r$ when the gradual decay on this parameter is implemented. Our experiments have consistently found that this configuration on $r$ causes a large scale discovery of the $\mathcal{S}$ region early in the early stages of the search. The *exploration-exploitation trade-off* in our algorithm is determined by the interplay between the trials used for each ECI optimisation, the quantity of samples obtained through the Rank-based sampling strategy, and their prioritisation level. An increase in the number of ECI optimisation trials improves the accuracy of policy optimisation but strains the inference time efficiency of the surrogate model. Then, a higher degree of prioritisation leads to an increase in sample efficiency but diminishes exploration, consequently reducing the potential for discovery of the satisfactory region. However, the issue of low exploration can be mitigated by increasing the number of Rank-based samples. Nonetheless, adding more samples per iteration results in slower re-training of the surrogate model in each iteration. Therefore, utilising probabilistic surrogate models capable of scaling to larger datasets and with faster inference time efficiency, such as Bayesian Neural Networks (BNNs) [61],

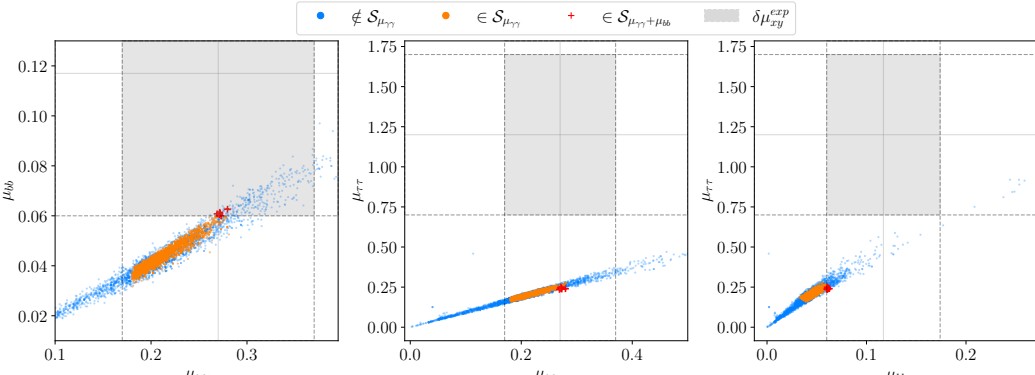

**Figure 9**: Scatter plots of the data set derived from the five-dimensional b-CASTOR search, evaluated in MG for $\mu_{\gamma\gamma}$, $\mu_{bb}$ and $\mu_{\tau\tau}$. It illustrates that while none of the points satisfy the three experimental signal-strengths modifier values simultaneously, nine points (marked with red crosses) meet the criteria for both $\mu_{\gamma\gamma}^{exp}$ and $\mu_{bb}^{exp}$, suggesting potential areas of interest for further exploration.

represents a potential avenue for further development.

We specifically compare our approach with MCMC-MH, as the latter is well-established in the community. Nevertheless, it is important to note that this comparison might not be entirely fair [19]. MCMC-MH operates under assumptions that we are testing against, such as requiring a long enough Markov Chain so that the MH algorithm samples the full posterior probability density function across the entire parameter space. This comparison serves as a proof-of-concept that a *search* algorithm could better suit our objectives of comprehensively characterising the $\mathcal{S}$ region in a sample-efficient manner. Exploring b-CASTOR performance against emerging approaches from the ML and AI community within HEP [21, 22, 60, 62] could yield valuable insights into potential enhancements and application scenarios.

## A   The TPE Algorithm

The Tree-structured Parzen Estimator (TPE) algorithm a variant of BO methods [49], commonly use for hyper-parameter optimisation in ML. In each iteration of our search we optimise ECI with TPE to generate a set of candidates. TPE utilizes Parzen Estimators as surrogate models to directly approximate $p(\boldsymbol{x} \mid y)$, formulated as follows:

$$p(\boldsymbol{x} \mid y) = \begin{cases} l(\boldsymbol{x}) & \text{if } y < y^* \\ g(\boldsymbol{x}) & \text{if } y \geq y^*, \end{cases} \tag{A.1}$$

where $l(\boldsymbol{x})$ and $g(\boldsymbol{x})$ are the probability densities modeling the two group of observations. The value $y^*$ is defined to be a quantile $\gamma$ of the observed $y$ values satisfying $p(y < y^*) = \gamma$. In TPE, $p(\boldsymbol{x}, y)$ is parameterised as $p(y)p(\boldsymbol{x} \mid y)$ to facilitate the optimisation of Expected Improvement (EI) acquisition function, although an explicit model for $p(y)$ is not needed since with this considerations the EI acquisition function becomes proportional to,

$$EI_{y^*}(x) \propto \left( \gamma + \frac{g(x)}{l(x)}(1 - \gamma) \right)^{-1} \tag{A.2}$$

Maximizing this equation leads to selecting points $x$ that predominantly align under the distribution $l(\boldsymbol{x})$ rather than $g(\boldsymbol{x})$.

## Acknowledgments

The work of SM is supported in part through the NExT Institute and the STFC Consolidated Grant No. ST/ L000296/1. MAD and his work were supported by ANID BECAS DE DOCTORADO EN EL EXTRANJERO, BECAS CHILE 2020, 72210042.

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
