# Peer review of "Bayesian Active Search on Parameter Space: a 95 GeV Spin-0 Resonance in the ($B-L$)SSM"

_SciPost Physics Core, doi:SciPost Phys. Core 8, 068 (2025)_

## Round 1 · Referee Report · Anonymous (Referee 1) · 2025-7-9

Strengths

1 - novel search algorithm is introduced in a clear way
2 - algorithm significantly outperforms the benchmark
3 - associated code is public and well documented

Weaknesses

1 - conclusions do not discuss further physics applications
2 - public code is not linked in the paper

Report

The authors present the novel search algorithm "b-CASTOR" for identifying valid parameter points of a BSM model. This is benchmarked against a Monte Carlo Markov Chain algorithm (MCMC-MH) on a simple test function and on Higgs bosons in the (B-L)SSM, aiming to explain excesses at 95 GeV. The two benchmarks show that b-CASTOR is significantly more efficient than MCMC-MH, as well as better able to identify distinct regions of validity.

The paper is well written. The physics application is explained in a self contained manner. The algorithm is introduced with a clear notation. A Python implementation is publicly available in a GitHub repository, which is useful and well documented; unfortunately it is not linked in the paper. The multidimensional results are displayed in well-designed figures which clearly demonstrate the efficiency of the new algorithm. The conclusions explore avenues of improving on b-CASTOR and the benchmarks. This could be extended with an outlook of which BSM models could most benefit from b-CASTOR.

The paper provides a valuable contribution to the task of identifying valid regions in parameter space of BSM models. Its methods can be applied to a range of models and the authors give an outlook for improving the algorithm. I therefore recommend to accept this paper for publication in SciPost after a minor revision (see "Requested Changes").

Requested changes

1 - In Section 2.3, second paragraph, the authors state that they use HiggsBounds and HiggsSignals to test for experimental constraints. Additional constraints could be obtained from recasting tools like MadAnalysis5. The authors should explain why they expect its contributions to be subleading or - if MadAnalysis5 was excluded because of runtime - state this as a limitation.
2 - The title of Section 3.4 "Performance study" seems unfitting since no study is performed in this section. Perhaps "Benchmark algorithm" or something similar might be clearer.
3 - The public code to reproduce the paper's results should be linked in Section 3.5. That is both the repo mjadiaz/blssm-bcastor and the implementation of bcastor in mjadiaz/hep-aid.
4 - It seems that the algorithm presented in the paper is applicable to a vast range of BSM models. This should be stated in the conclusions. It would be particularly useful if the authors could highlight BSM models that can most benefit from a search with b-CASTOR.

Typos:
5 - Section 1, paragraph "This work aims at filling...", first sentence change "being out benchmark" to "being our benchmark"
6 - Section 2.2, first paragraph change "constants are denoted as $y_v$" to "constants are denoted as $y_\nu$"
7 - Section 2.2, second paragraph last sentence change $SU(2)_I$ to $SU(2)_L$
8 - Section 4.2, first paragraph last sentence change "In this work, for we discard" to "In this work, we discard"

Recommendation

Ask for minor revision

  • validity: -
  • significance: -
  • originality: -
  • clarity: -
  • formatting: -
  • grammar: -

Author:  Mauricio A. Diaz  on 2025-09-16  [id 5830]

(in reply to Report 1 on 2025-07-09)

We sincerely thank the Editor and both referees for their careful reading and constructive feed-back. We have revised the manuscript accordingly and made relevant changes for a resubmission to a different journal option on SciPost, following the Editor's recommendation.

---

## Round 1 · Referee Report · Anonymous (Referee 2) · 2025-7-11

Report

See the attached file.

Attachment

Recommendation

Ask for major revision

  • validity: low
  • significance: good
  • originality: high
  • clarity: good
  • formatting: excellent
  • grammar: perfect

Author:  Mauricio A. Diaz  on 2025-09-16  [id 5829]

(in reply to Report 2 on 2025-07-11)

We sincerely thank the Editor and both referees for their careful reading and constructive feed-back. We have revised the manuscript accordingly and made relevant changes for a resubmission to a different journal option on SciPost, following the Editor's recommendation.

---

## Round 2 · Author Response

Response to Referees

We sincerely thank the Editor and both referees for their careful reading and constructive feedback. We have revised the manuscript accordingly and provide detailed responses below, quoting each comment as bullet points followed by our reply. We are grateful for the insightful suggestions, which have improved the clarity and scope of the paper.

Referee 1

  • While the comparison to traditional MCMC methods is appreciated, the work would benefit significantly from benchmarking against more modern scanning techniques such as those implemented in the BSMArt package (refs. [22,23]), which employs active learning for efficient exploration of BSM parameter spaces. A head-to-head comparison in terms of performance and computational efficiency would be particularly valuable for potential users seeking practical guidance on method selection.

Response: We appreciate this suggestion to include a comparison with modern parameter-scanning techniques such as those in the BSMArt package. As noted in our conclusions, benchmarking b-CASTOR against emerging ML/AI-based approaches is indeed a valuable direction for future work. A direct, fair comparison is not entirely straightforward because these algorithms often pursue different objectives: for example, the active-learning strategies in BSMArt aim to identify the boundaries of satisfactory regions using deep learning, whereas our primary objective is to densely populate the satisfactory region in a sample-efficient manner using Gaussian Processes.

We face a similar challenge even in our current comparison with MCMC--MH, which is fundamentally a sampling algorithm rather than a search algorithm. Our comparison to MCMC-MH is therefore intended as a proof of concept, demonstrating how a search-based approach like b-CASTOR can more efficiently characterise the satisfactory region.

Nevertheless, we agree on the importance of broader benchmarking and now emphasise in the conclusions that exploring b-CASTOR's performance relative to ML/AI-driven methods will be an important focus of future work.

  • In this context, including detailed and quantitative timing benchmarks with a breakdown of the computational cost per step, is essential. Timing results should also indicate the computing environment used (e.g. CPU model, GPU availability, memory), as such information is crucial for reproducibility and fair comparisons.

Response: We thank the referee for highlighting the importance of reproducibility. We have added a complete description of the hardware and computational setup used for our studies, together with timing information and a breakdown of costs per step.

  • Turning to the physics application, the authors choose a supersymmetric model motivated by dark matter, neutrino masses and anomalies pointing to the presence of a new states with a mass of 95 GeV. Given these motivations, it would be appropriate to include constraints relevant to all these phenomena, such as those from the dark matter relic abundance and direct/indirect detection constraints, neutrino observables, electroweak precision tests and LHC searches. At present, the study appears superficial: the scan focuses primarily on a subset of scalar states and the broader phenomenological implications are left unexplored. If the authors opt to present only a limited physics application as currently done, it would arguably be more appropriate to adopt a simplified model framework rather than a complex supersymmetric scenario. Moreover, the treatment of experimental uncertainties in the objective function remains unclear.

Response: We thank the referee for this thorough and constructive analysis. As suggested by the Editor, we are resubmit the work to \emph{SciPost Core}, which is better aligned with the scope of our study. We emphasise that the primary aim of this paper is not to provide an exhaustive phenomenological analysis, but rather to demonstrate the potential of b-CASTOR as a sample-efficient search strategy for complex BSM models. To make this clearer, we have expanded the introduction to explain our choice of the $(B-L)$SSM as a computationally challenging test case and to note explicitly that the algorithm is flexible enough to incorporate additional phenomenological constraints, such as those from dark matter, neutrino observables, and precision tests, in future work. We have also clarified how experimental uncertainties are incorporated into the objective function by explicitly stating the likelihood form and the associated bounds used.

  • Another important concern pertains to the method’s coverage of parameter space. In the shown examples, the authors note that viable regions may be missed (as seen in figure 3(c)). Although they mention that the algorithm can be guided to avoid such pitfalls, this feature appears not to be activated by default. It is thus not clear why a mechanism to mitigate this issue is not implemented systematically. This indeed raises concerns regarding the robustness of the method, particularly in the B$-$L supersymmetric scan where unexplored viable regions could remain. Somehow, without such proper safeguards, there is a risk that the algorithm may become trapped in local minima and fail to explore the full viable space.

Response: We thank the referee for raising this point. We have expanded the final paragraph of Section 4.1 to explain the theoretical guarantees of the ECI framework, which our batched algorithm preserves, ensuring asymptotic coverage of the satisfactory region as the number of samples increases. We use a decaying-radius strategy to balance early discovery with late-stage resolution, and we performed a grid search to select optimal $(r_{\mathrm{initial}}, r_{\mathrm{final}})$ values. We clarify when and how these safeguards are activated in practice.

  • The choice of references in the introduction is unusual and should be revised. For instance, the discussion of anomalies (refs. [3,4]) cites theoretical papers instead of the original experimental studies. Standard supersymmetry references are omitted (in favour of self-citations). Furthermore, the list of references related to the 95 GeV excess (refs. [9-12]) is incomplete and should be expanded to reflect the broader literature.

Response: We have added standard supersymmetry references and expanded the literature list on the 95 GeV excess. References [3,4] are retained as review-style summaries of anomalies, while original experimental studies are cited for the corresponding measurements where appropriate.

  • The manuscript introduces a large number of acronyms and abbreviations, including several that are uncommon. This hampers readability. The authors are encouraged to minimise the use of non-standard abbreviations.

Response: We have reduced the number of acronyms, removing, e.g., BMO (Bayesian Multi-Objective Optimisation) and AL (Active Learning), and we define remaining abbreviations at first use.

  • The sentence "The computational cost associated with numerically evaluating a specific configuration of a BSM model using a typical High Energy Physics (HEP) software toolbox is high." would benefit from a more precise and quantitative statement.

Response: We added the following quantitative example: "As an example, the HEP toolbox used in this study requires approximately one minute to evaluate a single parameter-space configuration".

  • In the introduction, it would be useful to cite representative works that rely on MCMC methods for BSM parameter space exploration, to better contextualise the need for alternative approaches.

Response: We have added representative examples, including HEPfit (which internally uses Metropolis-Hastings MCMC) and \texttt{emcee} (a parallel and more advanced MCMC sampler) with applications to BSM fits and dark-matter phenomenology.

  • Section 2.2 aims to briefly describe the model under consideration, but the current summary is overly concise. Reliance on external references makes the section insufficiently self-contained. At a minimum, the authors should include a table listing the quantum numbers of all superfields, as well as the supersymmetry-breaking Lagrangian. This would greatly aid the reader in understanding the scalar mass matrices discussed later as many parameters appear without being defined beforehand. Finally, note that the $\chi_2$ superfield is never introduced explicitly.

Response: We have defined all terms and parameters in Section 2.2, introduced the $\chi_2$ superfield, and clarified the Higgs mass-matrix parameter definitions. We have not added a full table of quantum numbers, as we believe this is not essential for our algorithm-focused presentation; however, the section is now self-contained for our purposes.

  • A reference to the HiggsTools framework should be included, alongside HiggsBounds and HiggsSignals.

Response: A reference to HiggsTools has been added alongside HiggsBounds and HiggsSignals.

  • It would be helpful to include information on how to access the code developed for this study (with a link to the associated public repository).}\end{quote}

Response: We have added links to both the \texttt{mjadiaz/blssm-bcastor} repository and the \texttt{mjadiaz/hep-aid} implementation.

Referee 2

In Section 2.3, second paragraph, the authors state that they use HiggsBounds and HiggsSignals to test for experimental constraints. Additional constraints could be obtained from recasting tools like MadAnalysis5. The authors should explain why they expect its contributions to be subleading or---if MadAnalysis5 was excluded because of runtime---state this as a limitation. Response: We thank the referee for this helpful suggestion. We have clarified in Sec. 2.3 that, for the 95 GeV anomaly, the experimental inputs we use are limited to the resonance mass and inclusive rates, which are directly constrained by HiggsBounds and HiggsSignals. We do not employ further event-level information, as such an approach would be necessary only for more fine-grained kinematic analyses; we have stated this explicitly as a limitation.

  • The title of Section 3.4 Performance study'' seems unfitting since no study is performed in this section. PerhapsBenchmark Algorithm'' or something similar might be clearer.

Response: The section title has been changed from "Performance study'' to "Benchmark algorithm".

  • The public code to reproduce the paper’s results should be linked in Section 3.5. That is both the repo \texttt{mjadiaz/blssm-bcastor} and the implementation of bcastor in \texttt{mjadiaz/hep-aid}.

Response: We have added links to both the mjadiaz/blssm-bcastor repository and the mjadiaz/hep-aid implementation.

  • It seems that the algorithm presented in the paper is applicable to a vast range of BSM models. This should be stated in the conclusions. It would be particularly useful if the authors could highlight BSM models that can most benefit from a search with b-CASTOR.

Response: We thank the referee for this comment and fully agree that the algorithm has broader applicability. We have extended the conclusions to state explicitly that b-CASTOR can be applied to a wide range of BSM scenarios. In addition to our detailed study of the $(B-L)$SSM, we now also mention an internal cross-check of its performance in the 2HDM, which showed a comparable level of efficiency.

Typos: We thank the referee for the careful list of corrections. All typos have been fixed.

Once again, we thank the Editor and both referees for their constructive feedback and guidance. We hope the revisions address all concerns satisfactorily and improve the clarity and usefulness of the manuscript.

We therefore ask for the paper to be reconsidered for publication.

Regards, The authors

---

## Round 2 · List of Changes

1) Page 2, paragraph 2:
- Expanded 95 GeV excess literature list and added standard SUSY references

2) Page 2, paragraph 4:
- Added representative works using MCMC (HEPfit, emcee, etc.)

3) Page 2, footnote 2:
- Added quantitative timing example

4) Page 3, paragraph 2 and footnote 4:
- Added links to mjadiaz/blssm-bcastor and mjadiaz/hep-aid

5) Page 4, paragraph 1:
- Expanded introduction to justify using the (B-L)SSM as a computationally challenging case

6) Section 2.2:
- Improved model description and defined all terms and parameters

7) Section 2.3, paragraph 2:
- Clarified why only HiggsBounds/HiggsSignals were used
- Stated exclusion of MadAnalysis5 as a limitation

8) Section 4.1, last paragraph on page 15:
- Emphasised that the batched extension preserves ECI’s convergence guarantees

9) Section 3.4 title:
- Changed title from "Performance study" to "Benchmark algorithm"

10) Section 4.3 (Computational Resources):
- Added computing environment details (hardware, CPU/GPU, memory)
- Added quantitative timing results and breakdown of computational costs per step

11) Conclusions:
- Extended to state b-CASTOR applies to a wide range of BSM models, mentioning internal cross-check with 2HDM
- Emphasised that comparisons with ML/AI-based methods are left for future work, noting that differing objectives make direct comparisons non-trivial

---

## Editorial Decision

published